# The Effects of Fermented Feed on the Growth Performance, Antioxidant Activity, Immune Function, Intestinal Digestive Enzyme Activity, Morphology, and Microflora of Yellow-Feather Chickens

**DOI:** 10.3390/ani13223545

**Published:** 2023-11-16

**Authors:** Fei Xu, Hongzhi Wu, Jiajun Xie, Tao Zeng, Lijian Hao, Wenwu Xu, Lizhi Lu

**Affiliations:** 1State Key Laboratory for Managing Biotic and Chemical Threats to the Quality and Safety of Agro-Products, Institute of Animal Husbandry and Veterinary Science, Zhejiang Academy of Agricultural Sciences, Hangzhou 310002, China; 2Junan Agriculture and Rural Bureau, Linyi 276600, China; 3Tropical Crop Genetic Resource Research Institute, Chinese Academy of Tropical Agricultural Sciences, Haikou 571101, China; 4Laboratory of Livestock and Poultry Resources (Poultry) Evaluation and Utilization, Ministry of Agriculture and Rural Affairs of China, Hangzhou 310002, China

**Keywords:** fermented feed, yellow-feather chickens, antioxidant activity immune function, intestinal health

## Abstract

**Simple Summary:**

The chicken digestive system is not fully developed, and the feed type and shape have essential effects on it. The growth performance and health status would be affected in various ways during this stage. Immune function and antioxidant capacity can explain this mechanism at the endocrine level. Fermented feed is a new feed type that saves food crops, has the potential to promote growth and development, improve animal welfare and help them stay healthy, and is beneficial to human food safety. We studied the effects of fermented feed on the growth performance, antioxidant activity, immune function, intestinal digestive enzyme activity, morphology, and microflora of yellow-feather chickens. The results showed that adding fermented feed increased the digestive enzyme activity, ameliorated intestinal morphology, cecal microflora, beneficial bacteria richness, immune function, and antioxidational ability of chickens without effects on growth performance. In conclusion, the fermented feed added to the chickens’ diets improved the growth performance, antioxidant activity, immune function, intestinal digestive enzyme activity, morphology, and microflora of yellow-feather chickens. This study offers a more theoretical basis for exploiting and utilizing new fermented feed resources and the sustainable and healthy development of poultry farming.

**Abstract:**

This experiment was conducted to investigate the effects of fermented feed on growth performance, antioxidant activity, immune function, intestinal digestive enzyme activity, morphology, and microflora of yellow-feather chickens. A total of 240 one-day-old female yellow-feathered (Hexi dwarf) chickens were randomly divided into two treatment groups, with six replicates per group and 20 chickens per replicate. The control group (CK) received a basal diet, whereas the experimental group was fed a basal diet of +2.00% fermented feed (FJ). The trial lasted for 22 days. Compared with the CK, (1) the growth performance was not affected (*p* > 0.05); (2) immunoglobin a, immunoglobin g, immunoglobin m, interleukin-1β, and interleukin-6 were affected (*p* < 0.05); (3) liver superoxide dismutase, glutathione peroxidase, and catalase were higher (*p* < 0.05); (4) trypsin activity in the duodenum and cecal Shannon index were increased (*p* < 0.05); (5) the relative abundance of *Actinobacteriota* in cecum was increased (*p* < 0.05); (6) the abundance of dominant microflora of *Bacteroides* as well as *Clostridia UCG-014_norank* were increased (*p* < 0.05). In summary, the fermented feed improved the growth performance, antioxidant activity, immune function, intestinal digestive enzyme activity, morphology, and microflora of yellow-feather chickens.

## 1. Introduction

As chickens’ digestive system and immune systems are poorly developed, the feed utilization rate and disease resistance are low [1,2,3]. The feed contains a large number of anti-nutritional factors, like cellulose, hemicellulose, and pectin, that affect the chicken digestibility [4] and have a specific impact on the survival rate of chickens. Currently, there is a shortage of feed resources and low utilization of agricultural by-products for other products. Therefore, it is imperative to improve the feed conversion rate, solve the shortage of resources, and utilize the by-products to reduce industrial waste [5]. Fermented feed refers to the high-quality feed produced by the mixed probiotic fermentation using different feed raw materials, auxiliary materials, and probiotics under specific temperatures, humidity, oxygen content, and pH, which can not only reduce anti-nutritional factors in the feed but also improve the content of beneficial bacteria, amino acids, short-chain fatty acids, peptides, and other nutrients [6]. Fermented feed has a promising future in promoting animal growth performance and immunity, replacing antibiotics, realizing the reuse of industrial waste, and alleviating the shortage of feed resources as a new type of feed with safe, green, non-toxic, and replaceable antibiotics [7,8,9]. There have been 44 bacteria strains announced by the Food and Drug Administration and the Association of American Feed Control Officials that can be used in fermented feed production. In recent years, the rapid development of fermented products has had a significantly good effect on animal husbandry production [10,11,12].

Growth performance is an essential and direct index for measuring the economic benefits and competitiveness of farming. The lactic acid bacteria increase the palatability of poultry feed [13] and are used as natural preservatives in animal feed to prevent fungal growth followed by mycotoxin production [14]. Li et al. [15] and Sun et al. [16] found that fermented feed could improve the average daily feed intake and average daily gain of broilers. Antioxidant capacity and immune function are closely related to animal health, indirectly affecting the economic benefits of breeding. It was shown that chickens supplemented with fermented feed would perform better in muscle antioxidant capacity [17]. More polypeptides, fatty acids, nitrogen-free extract, and other nutrients are produced during the fermentation process by certain feed resources, which provide material and energy for immunity [18,19,20], and nutrition is the foundation of immunity [21]. However, few studies have been conducted to comprehensively evaluate the growth, immunity, antioxidant, digestion, intestine morphology, and microflora of chickens. This study aims to investigate the effects of fermented feed on the growth performance, antioxidant activity, immune function, intestinal digestive enzyme activity, morphology, and microflora of yellow-feather chickens and offer a more theoretical basis for the exploitation and utilization of new fermented feed resources and the sustainable and healthy development of poultry farming.

## 2. Materials and Methods

### 2.1. Experiment Material

Fermented feed is made by mixing beancurd residue and beer grains with bran and rice hull powder, followed by adding lactic acid bacteria and yeast (both the number of viable bacteria is 2.0 × 10^7^ CFU/g) and then fermenting in 20 kg sealed plastic bags at 32 °C, humidity 67.0%, and pH 4.2–4.3 for 11 days. The basal diet and fermented feed used in the experiment were purchased from Jiangxi Qiling Agriculture and Animal Husbandry Co., Ltd., Ganzhou, China. All the experimental chickens received 24 h light every day and were raised on the floor, had free access to fresh drinking water, were fed twice a day at 8:00 and 17:00, respectively, and were vaccinated according to the normal immunization procedures. Automatic temperature control equipment was used to control the temperature of the hen house at 30 °C. The mental state, the disease, and the death of the chickens were observed and recorded. The experimental animals were offered by Qiling Agriculture and Animal Husbandry Farm, Ganzhou, China.

### 2.2. Experiment Design and Sample Collection

A total of 240 one-day-old female yellow-feathered (Hexi dwarf) chickens with similar body weights were randomly divided into two groups, with six replicates per group and 20 broilers per replicate. The control group (CK) was fed a basal diet, and the fermented feed group (FJ) was fed a basal diet of +2.00% fermented feed, determined based on previous experiments, and the experimental basal diet was corn–soybean diet, which was formulated according to the nutrient requirements published by NRC (1998; 2012). Basal diets, fermented feed composition, and nutrient levels are shown in Table 1. The trial lasted for 22 days.

Body weights were measured for each chicken at 1 and 22 d of age, and the total feed consumption of each group was recorded. Growth performance was evaluated by average daily gain (ADG), average daily feed intake (ADFI), and feed conversion rate (FCR). At the end of the experiment, one chicken that was similar in average weight was selected from each replicate to be slaughtered by carotid artery bloodletting after feed deprivation for 12 h, and about 10 mL of blood samples was collected from the wing artery of each replicate in a tube before the chicken was slaughtered. Then, the abdominal cavity was rapidly opened, and the contents were taken, respectively—about 2 g from the duodenum, jejunum, ileum cecum, and liver, which were placed in a sterile cryogenic vial and stored at −80 °C for later use. Tubular segments (2–3 cm long) in the same part were cut from the duodenum, jejunum, and ileum; cleaned with normal saline; and stored in 4% paraformaldehyde fix solution at room temperature. The serum was separated by centrifugation for 20 min at 3500 r/min at 4 °C and stored at −20 °C for later analysis.

The immunity indexes contain immunoglobin A (IgA), immunoglobin G (IgG), immunoglobin M (IgM), interleukin-1β (IL-1β), and interleukin-6 (IL-6). The antioxidant indexes contain malonaldehyde (MDA), superoxide dismutase (SOD), glutathione peroxidase (GSH-Px), and catalase (CAT). They were all determined by the HITACHI Automatic Analyzer 3500, Ibaraki-Ken, Japan, and the kits were provided by Beijing SINO-UK Institute of Biological Technology. The catalog numbers for IgA, IgG, IgM, IL-1β, IL-6, MDA, SOD, GSH-Px, and CAT kits are ILP2673-C1, 269A-16-RUO, M2521, EP2074, EP2076, A003-1, N82750-25g-, S10152-200UN-, and A007-1, respectively.

The duodenum, jejunum, and ileum contents were taken at 0.3 g, respectively, and normal saline was added to the mass ratio of the contents: normal saline = 1:9, treated with a high-speed grinder under the ice bath condition, centrifuged at 3000 r/min for 10 min at 4 °C with Multifuge X4 Pro ThermoFisher High-performance centrifuge (Shanghai, China), and 50 μL of supernatant was taken to determine the amylase, trypsin, and lipase activities. The kit was provided by Nanjing Jiancheng Bioengineering Co., Ltd., Nanjing, China. The catalog numbers for the amylase, trypsin, and lipase kits are Ab102523, PP0100-1KT, and Ab102524, respectively.

The duodenum, jejunum, and ileum tissues were fixed with 4% paraformaldehyde fix solution for 72 h, and then paraffin sections were prepared, and hematoxylin–eosin (HE) staining was performed. A 40-fold vision of view of scan images was captured by K-Viewer software (1.7.0.29) for observation. Six tissue regions with complete intestinal morphology and a clear field of view were selected for scanning. Using a micron as the standard unit, Image-pro plus 6.0 was used to measure the villus height and crypt depth of a single intestine, and the villus height/crypt depth was calculated.

The cecum contents were taken from the −80 °C refrigerator, and its total DNA was extracted using E.Z.N.A.^®^ Stool DNA Kit (catalog number: YFXM0027), which was provided by Shanghai Lingen Biological Technology Co., Ltd., Shanghai, China. The purity and concentration of DNA extraction were detected by 1% agarose gel electrophoresis. According to the total DNA of cecum contents as a template, the rRNA gene V3–V4 region was amplified by PCR using high-fidelity DNA polymerase. The upstream primer was 515F (5′-barcode-GTGCCAGCMGCCGCGG)-3′), and the downstream primer was 907R (5-CCGTCAATTCMTTTRAGTTT-3′) for PCR amplification. The library was established, followed by the 16S rDNA sequencing on an Illumina HiSeq 2500 PE 250 platform in Shanghai Biozeron Biotechnology Co., Ltd., Shanghai, China. All the sequence splicing and filtering were performed using the fastqc software. Qiime software (Version 1.7.0) was used for OUT (Operational Taxonomic Units) clustering and statistical analysis in bioinformatics at the 97% similarity level. Based on the clustering results, α diversity index analysis, β diversity index analysis, principal component analysis, microbial classification statistics, and linear discriminant analysis effect size (lEfSe) were conducted. Statistical analysis of community structure was performed at taxonomic levels (phylum, class, order, family, genus).

### 2.3. Statistical Analysis of Data

Statistical analyses were conducted using SPSS 20.0 statistics software. Different treatments were analyzed using two-factor and multiple comparisons with Tukey’s multiple-range tests. Data were expressed as mean ± standard error of the mean and *p* < 0.05 as the significance differences.

## 3. Results

### 3.1. Effects of Fermented Feed on Growth Performance in Chicks

The differences in ADG, ADFI, and FCR of chicks between CK and FJ were not statistically significant (*p* > 0.05) (Figure 1).

### 3.2. Effects of Fermented Feed on Serum Immune Function of Chickens

The IgA, IgG, and IgM were significantly affected (*p* < 0.01) by the fermented feed. The IL-1β and IL-6 were lower (*p* < 0.01) in FJ than those in CK (Table 2).

### 3.3. Effects of Fermented Feed on Antioxidational Ability in the Liver of Chickens

The differences in MDA, SOD, GSH-Px, and CAT in the liver of chickens between CK and FJ were statistically significant (*p* < 0.05). The liver MDA in FJ was lower than that in CK; however, the liver SOD, GSH-Px, and CAT of CK were higher than that of FJ (Figure 2).

### 3.4. Effects of Fermented Feed on Digestive Enzyme Activity in Chickens

Compared with CK, the duodenum trypsin activity in FJ was significantly enhanced (*p* < 0.05), and there were no statistically significant differences (*p* > 0.05) with the activities of neither trypsin in the jejunum and ileum nor amylase together with lipase in the duodenum, jejunum, and ileum in FJ (Table 3).

### 3.5. Effects of Fermented Feed on Intestinal Tissue Morphology of Chickens

The villus height, crypt depth, and height/crypt depth among the two groups did not differ significantly (*p* > 0.05), except the villus height of ileum in FJ was significantly lower (*p* < 0.01) than CK (Table 4).

### 3.6. Effects of Fermented Feed on Microbial Diversity in the Cecum of Chickens

#### 3.6.1. Alpha Diversity Analysis

Compared with CK, the Shannon indexes in FJ were significantly increased (*p* < 0.05). However, the differences between all the Chao indexes, Richness indexes, ACE indexes, Evenness indexes, and Simpson indexes were not significant (*p* > 0.05) (Table 5).

#### 3.6.2. Beta Diversity Comparative Analysis–PCA Analysis of the Chicken Samples between the Groups

As shown in Figure 3, the microbial colony structure difference between CK and FJ was significant, indicating that the addition of fermented feed had a significant influence (*p* < 0.05) on the intestinal flora structure of chickens’ cecum.

#### 3.6.3. Relative Abundance of Cecal Microflora at Phylum and Genus Levels of Chickens

Based on the analysis at the phylum level, it can be seen that the dominant microflora in cecum contents was mainly *Bacteroidota*, *Firmicutes*, *Actinobateriota*, *Proteobacteria,* and *Desulfobacterota. Firmicutes* was the main bacteria in the cecum intestinal flora. Compared with CK, the *Desulfobacterota* was extremely significantly increased (*p* < 0.01), and *Actinobateriota* was significantly increased (*p* < 0.05), while the differences of *Bacteroidota*, *Firmicutes,* and *Proteobacteria* were not significant (*p* > 0.05) in FJ (Figure 4).

The figure that was carried out by the analysis at the genus level indicated the dominant flora of intestinal contents bacteria in cecum mainly included *Barnesiella*, *Bacteroides*, *Alistipes*, *Clostridia UCG-014_norank*, and *Faecalibacterium*. The highest two bacteria contents were *Barnesiella* and *Bacteroides*, the dominant fungi in the microecological environment of cecum. It can be seen after analysis that the dominant microflora abundance of *Bacteroides* and *Clostridia UCG-014_norank* were significantly increased (*p* < 0.05), and *Barnesiella* was significantly decreased (*p* < 0.05). At the same time, both *Alistipes* and *Faecalibacterium* were not significantly affected (*p* > 0.05) by FJ compared to CK (Figure 5).

#### 3.6.4. Lefse Difference Analysis of the Chicken Samples between the Groups

Only the species with LDA values greater than 2.4 were displayed in Figure 4, which were used to evaluate species groups with significant differences. There were 13 species groups with significant differences in FJ, which were mainly about *Clostridia_UCG_014*, *RF39*, *Pseudomonadales*, *Moraxellaceae*, *Acinetobacter*, *Bilophila*, *Puniceicoccaceae*, *Opitutales,* and so on. While *CHKCI001*, *Erysipelothrix*, *Negativibacillus*, *Sphingomonas*, *Sphingomonadales*, *Sphingomonadaceae,* and *Oscillospirales* were the leading nine significantly different species groups (Figure 6).

## 4. Discussion

### 4.1. Effects of Fermented Feed on the Growth Performance of Chickens

Various amino acids, peptides, organic acids, vitamins, and other nutrients are generated during the feed fermentation [6]. In addition, the anti-nutritional factor contents can be reduced by fermentation [22,23,24], thereby improving the decomposition, digestion, absorption, and utilization of nutrients in animals. There are not only chemical changes occurring in fermentation but also physical changes happening. The feed pH value, hardness, shape, and taste will be changed after fermentation [25,26]. Both chemical and physical changes could affect the feed palatability and the animal growth performance. Zhu et al. [26] found that the feed intake of chickens fed 10% and 15% dried or 10% wet fermented feed during the whole period (1–42 d) was higher than those fed the control diet, while feed intake and feed conversion ratio were not affected by those fermented feed during the starter stage (1–21 d). Omar et al. [27] found that adding fermented fava bean by-products of 15%, 25%, and 35% improved the feed intake of broiler chickens. Xie et al. [28] found that the weight gain increased significantly after 10% of the olive leaf residues fermented by solid-state fermentation were added to daily chicken feed for 28 days. This experiment showed that the chickens’ ADG, ADFI, and FCR were not affected by fermented feed, which may be due to some of the following reasons: (1) The chickens were fed in hot and damp weather, harming growth performance, as the weather induces heat stress, promoting catabolism in chickens [29] and decreasing the feed intake [30]. (2) The development of the digestive system is immature, making the absorption of nutrients incomplete. (3) The stomach content of chickens is limited, and the yellow-feather broilers have a large feed intake, which makes it easy to reach the maximum stomach content, and they cannot continue to increase feed intake.

### 4.2. Effects of Fermented Feed on Immune Function in the Serum of Chickens

Specific immunity mainly consists of humoral immunity and cellular immunity [31]. The immune system is mediated by cells, soluble factors, interacting cells, and tissues [32]. Immunity is the biological basis of organisms resisting pathogenic microorganisms, bacteria, and viruses. The function of immunity is mainly reflected by immune indicators. Immunoglobulin (Ig), which mainly includes IgG, IgM, and IgA, refers to any class of structurally related proteins in the serum and the immune system cells [33]. The systemic inflammatory response is triggered by some cytokines, including interleukin 1 (IL-1), IL-6, and tumor necrosis factor (TNF-α), which lead to multiple inflammatory cascades [34]. The immune system enhancement is illustrated by the decrease of the proinflammatory cytokines content. Zhu et al. [26] found that the serum IgA, IgG, and IgM of broilers in groups fed with 10%, 15%, and 25% dried fermented feed and 10% wet fermented feed were higher than the control group, except for IgG content in the 25% dried fermented feed group. Wang et al. [35] found that fermented soybean meal decreased the plasma IL-1β and IL-6 concentration of weaned piglets after the enterotoxigenic *Escherichia coli* (ETEC) K88 challenge. And several previous studies have suggested that fermentation could boost immunity [36,37,38]. The results of this experiment showed that the serum IgA, IgG, and IgM of chicks in FJ were significantly increased, while the levels of proinflammatory factors IL-1β and IL-6 in serum decreased significantly compared with the CK. The results showed that adding fermented feed during the brooding period could significantly improve the serum immunity function of chickens.

### 4.3. Effects of Fermented Feed on Antioxidational Ability of Chickens

The oxygen toxicity in living organisms mainly comes from the oxygen free radicals produced by its partial reduction, also known as reactive oxygen species (ROS), which are by-products of normal cellular metabolism [39]. It may lead to oxidative stress and damage to the cells and biomolecules when animal ROS is excessive [40]. Lipid and protein oxidations are generally initiated by ROS [41]. The ability of ROS to facilitate protein oxidation was enhanced with malondialdehyde (MDA) [42]. MDA is commonly used as an index of oxidative stress, which is highly reactive and disseminates and magnifies oxidative damage during lipid peroxidation [43]. Superoxide dismutase (SOD), glutathione peroxidase (GSH-Px), and catalase (CAT) belong to the first level of the antioxidant defense network [44]. Zhu et al. [45] reported that the T-AOC, SOD, and GSH-Px levels in the serum of chickens were markedly elevated by fermented feed, while the serum MDA level was significantly reduced by that. Zhang et al. [46] found that liver MDA in the control group was less than that in the Aspergillus niger fermentation group and Candida utilis combined with the Aspergillus niger fermentation group. Niu et al. [47] reported that fermented Ginkgo biloba leaves supplementing could improve the broiler gut’s antioxidant activity. In addition, the antioxidational ability was improved by fermented feed in different tissues and organs of animals. In this study, therefore, it is inferred that the liver antioxidational ability of chickens was promoted by adding fermented feed to the basal feed.

### 4.4. Effects of Fermented Feed on Intestinal Digestive Enzyme Activity of Chickens

Animal growth performance is determined by feed utilization, which is closely related to the digestion enzyme activity [27]. The digestive enzymes decompose the significant molecular substances into small ones, which the intestine absorbs more efficiently [48]. This chemical digestion can raise the feed utilization rate to improve animals’ growth performance [27,48]. Amylase is a critical enzyme in hydrolyzing starch and polysaccharides into disaccharides and oligosaccharides [49]. Proteins and peptide chains can be hydrolyzed by trypsin into easily absorbable polypeptides and amino acids [50]. Monoglycerides and free-fatty acids can be efficiently absorbed by organisms and decomposed from lipids by lipase [51]. Feng et al. [52] found that the addition of fermented soybean meal replacing soybean meal in 1–21-day-old broilers’ diet significantly increased trypsin, chymotrypsin, and lipase in the small intestine. It was found that replacing soybean meal with fermented cottonseed meal in the diet improved the amylase and protease activities in intestinal content after 21 days and also improved the protease activity in intestinal digesta of 42-day-old broilers [53]. It was shown that the *AMY2A* gene expression was significantly upregulated after feeding chickens with fermented fava bean by-products. In contrast, the *PNLIP* gene expression was significantly upregulated with increasing fermented fava bean levels [27]. The results demonstrated that the duodenal amylase activity of chickens could be significantly improved by adding fermented feed. In contrast, the duodenum amylase, jejunum, ileum amylase, trypase, and lipase were not significantly improved. Although the overall trend was increased, there were a few results with significant differences. The differences of multiple data sets may be caused by the short feeding period.

### 4.5. Effects of Fermented Feed on Intestinal Tissue Morphology of Chickens

The intestinal epithelium is covered by a superficial columnar epithelium layer and divided into crypt and villus. The crypt is invaginated into the hypoblast layer mesenchyme, and the villus is oriented towards the intestinal lumen [54]. It was found that nutrient absorption is mainly performed in the small intestine, and the villus height (VH), crypt depth (CD), and VH/CD determined the absorption efficiency of nutrients in diets [55]. Sun et al. [53] found that the broilers fed fermented cottonseed meal have higher VH and VH/CD in the jejuna and duodena. The feed fermentation creates organic acids, which stimulate gastro-intestinal cell proliferation and potentially increase the intestinal surface area by improving the VH [56,57]. The higher VH/CD indicated a more vital integrated functional state of the small intestine [58]. It was shown that the chicken fed high doses of fermented feed improved the villus height in the duodenum and a higher VH/CD in the duodenum and ileum [59]. The results of this experiment showed that the supplementation of fermented feed could reduce the crypt depth of the duodenum, jejunum, and ileum and increase the villus height as well as having higher VH/CD in the duodenum and jejunum. This suggests that fermented feed can improve intestinal structure, echoing the above results of improving the growth performance of broilers.

### 4.6. Effects of Supplementing Fermented Feed on Microbial Diversity in the Cecum of Chickens

The stability, resistance, and resilience of intestinal microflora are the most basic ecological characteristics that play an essential role in animal health, and it was estimated that 100 trillion bacteria from 500 to 1000 species exist in the intestine [60]. Trillions of commensal microbial flora constitute a microecology [61]. Intestinal microecology balance is inextricably linked with animals’ health level, immune ability, and growth performance [62]. Lv et al. found that fermented feed changed the diversity and richness of animal intestinal flora, which provided a better environment for the colonization and growth of beneficial bacteria [63], so we collected caecum contents of chickens for further exploration. It was reported that *Firmicutes* and *Bacteroidota* are the top two dominant bacterial phyla in poultry intestines [64,65]. The results of this study showed that *Bacteroidota* and *Firmicutes,* accounting for more than 90% of the total cecum microbiotas, were the dominant phylum microbiotas, which were similar to the previous study. *Bacteroidota* and *Firmicutes* participate in glycolysis and promote the absorption and metabolism of nutrients [66]. The dietary addition of fermented feed significantly increased the relative abundance of *Desulfobacterota* in the cecum of chickens, indicating that fermented feed promoted the colonization of *Desulfobacterota* in the cecum of chicks.

Nevertheless, *Desulfobacterota*, a sulfate-reducing phylum, is harmful to the intestine, which reduces sulfate to hydrogen sulfide [67]. The analysis results at the genus level showed that feeding fermented feed significantly increased the number of *Bacteroides*. This Gram-negative obligately anaerobic non-spore-former bacterium regulates intestinal flora’s microecological balance to facilitate intestinal flora’s colonization and participate in the degradation of polysaccharides generating acetate and propionate [68]. *Bacteroides* play a role in maintaining intestinal homeostasis [69] and transforming bile acids, thus improving the deposition rate of nutrients [70]. We found that adding fermented feed to diets significantly improved the relative abundance of *Bacteroides* and *Clostridia UCG-014_norank* in the cecum.

In contrast, the relative abundance of *Alistipes* and *Faecalibacterium* tended to increase with an insignificant difference, indicating that fermented feed improved the abundance of bacteria, promoted the growth of beneficial bacteria, and improved the fermentation function of cecum bacteria. *Faecalibacterium* are widely found in animal intestines, mainly fermented to produce short-chain fatty acids, which can inhibit pathogens producing pathogenic factors, provide energy for intestinal epithelial cells, and are widely used to develop probiotics and biological drugs. In summary, the fermented feed affected the composition of the recipient animals’ intestinal flora, which increased the proportion of beneficial bacteria such as *Actinobateriota*, *Bacteroides,* and *Clostridia UCG-014_norank*.

## 5. Conclusions

In this study, the 2.00% fermented feed improved the growth performance, antioxidant activity, immune function, intestinal digestive enzyme activity, morphology, and microflora of yellow-feather chickens.

## Figures and Tables

**Figure 1 animals-13-03545-f001:**
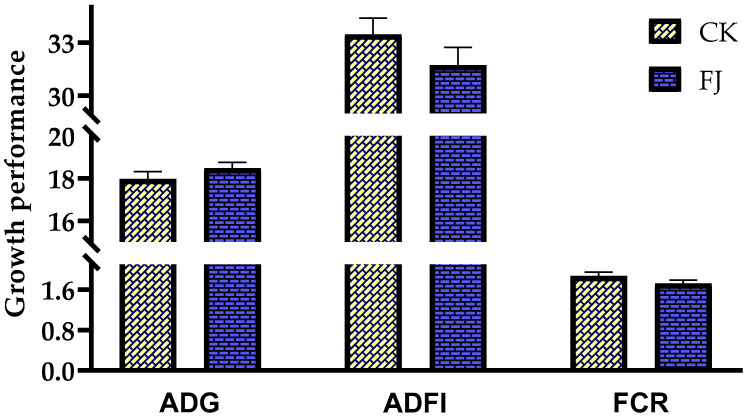
Effects of fermented feed on growth performance of chicks. ADG (Average daily gain) in the CK and FJ groups was 17.96 ± 0.36 vs. 18.47± 0.28, respectively, *p* = 0.288; ADFI (Average daily feed intake) in the CK and FJ groups was 33.46 ± 0.93 vs. 31.73 ± 1.00, respectively, *p* = 0.234; FCR (Feed conversion rate) in the CK and FJ groups was 1.87 ± 0.08 vs. 1.72 ± 0.07, respectively, *p* = 0.192.

**Figure 2 animals-13-03545-f002:**
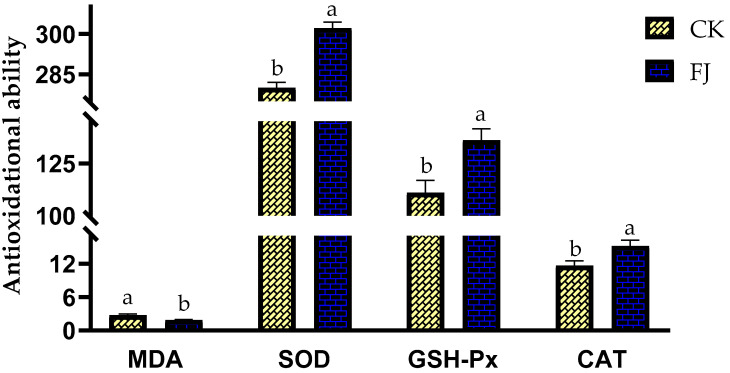
Effects of fermented feed on immune index in the liver of chickens at 22 d of age. Note: The different superscripts within the same index indicate a significant difference (*p* < 0.05). The MDA (Malonaldehyde) levels in the CK and FJ groups were 2.74 ± 0.24 ^a^ vs. 1.87 ± 0.12 ^b^, nmol/mg·protein, respectively, *p* = 0.009; The SOD (Superoxide dismutase) levels in the CK and FJ groups were 280 ± 2.06 ^b^ vs. 302 ± 2.33 ^a^, U/mg·protein, respectively, *p* = <0.001; the GSH-Px (Glutathione peroxidase) levels in the CK and FJ groups were 111 ± 5.92 ^b^ vs. 136 ± 5.47 ^a^ U/mg·protein, respectively, *p* = 0.010; The CAT (Catalase) levels in the CK and FJ groups were 11.67 ± 0.84 ^b^ vs. 15.19 ± 1.04 ^a^, U/mg·protein, respectively, *p* = 0.025.

**Figure 3 animals-13-03545-f003:**
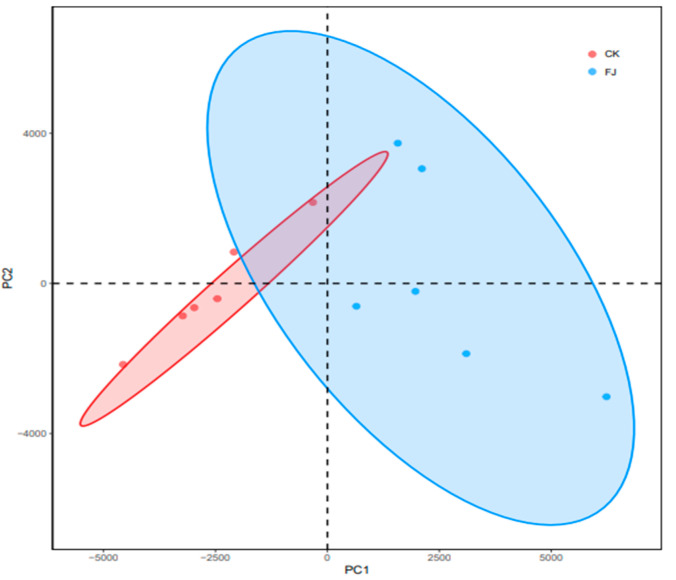
PCA analysis of microorganisms in cecal contents of chickens at 22 d of age.

**Figure 4 animals-13-03545-f004:**
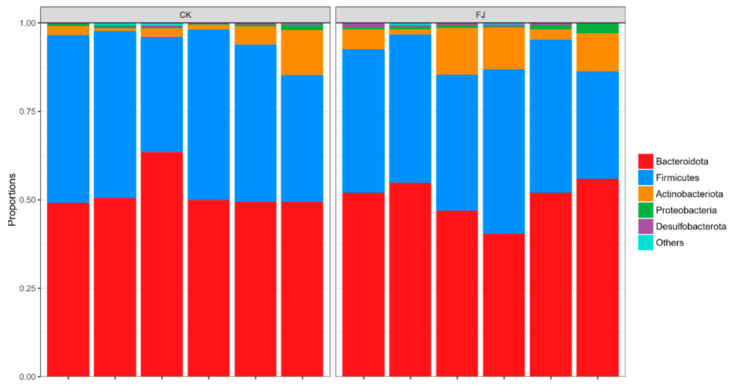
Relative abundance of cecal microflora at phylum level of chickens at 22 d of age.

**Figure 5 animals-13-03545-f005:**
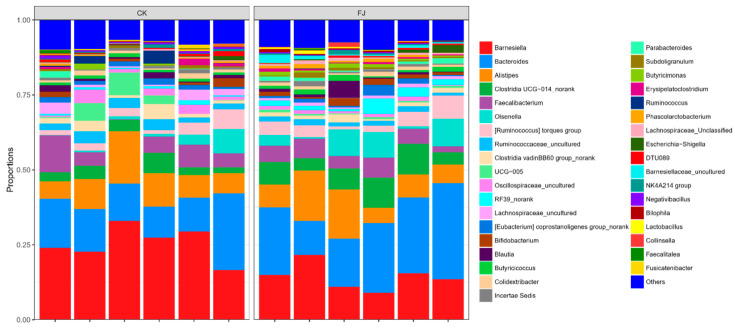
Relative abundance of cecal microflora at genus level of chickens at 22 d of age.

**Figure 6 animals-13-03545-f006:**
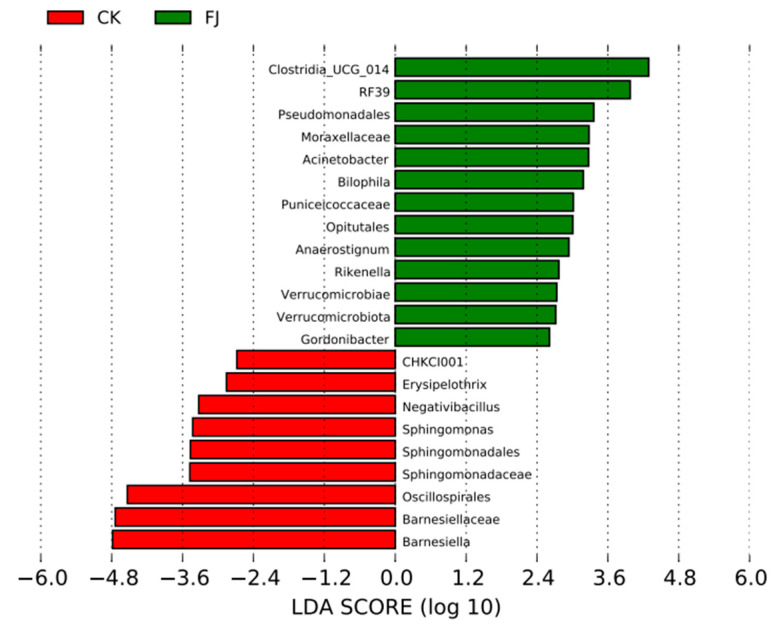
Column diagram of LefSe analysis of cecal contents in chickens at 22 d of age.

**Table 1 animals-13-03545-t001:** Composition (kg/100 kg) and nutrient level of the experimental diets ^1^ for chickens.

Basal Diets	Contents	Fermented Feed	Contents
Ingredients		Ingredients	
Corn, %	21.30	Corn,%	20.59
Wheat bran, %	43.64	Beancurd residue, %	30.12
Soybean meal, %	19.26	Beer grains, %	20.26
Extruded soybeans, %	3.00	Soybean meal, %	11.79
Sunflower seed meal, %	2.00	Extruded soybeans, %	3.50
Peanut meal, %	3.00	Sunflower seed meal, %	2.56
Corn gluten meal, %	2.56	Corn gluten meal, %	3.00
Lard oil, %	1.07	sunflower seed oil,%	4.01
Calcium hydrogen phosphate, %	1.09	Calcium hydrogen phosphate, %	1.09
Limestone, %	1.34	Limestone, %	1.34
Baking soda, %	0.20	Baking soda, %	0.20
Salt, %	0.25	Salt, %	0.25
Methionine (95%), %	0.23	Methionine (95%), %	0.23
Threonine (95%), %	0.15	Threonine (95%), %	0.15
Lysine sulphate, %	0.69	Lysine sulphate, %	0.69
Choline, %	0.08	Choline, %	0.08
Premix ^2^, %	0.14	Premix ^2^, %	0.14
Total	100.00	Total	100.00
Nutrient levels, on air-dry basis		Nutrient levels, on air-dry basis	
Metabolic energy ^3^, ME, MJ/kg	12.10	Metabolic energy ^3^, ME, MJ/kg	12.11
Crude protein ^4^, CP, %	21.00	Crude protein ^4^, CP, %	20.93
Crude fiber ^4^, CF, %	2.96	Crude fiber ^4^, CF, %	2.98
Calcium ^4^, Ca, %	0.88	Calcium ^4^, Ca, %	0.89
Phosphorus ^4^, P, %	0.60	Phosphorus ^4^, P, %	0.59
Lysine ^4^, Lys, %	1.15	Lysine ^4^, Lys, %	1.14
Methionine ^4^, Met, %	0.52	Methionine ^4^, Met, %	0.53

^1^ Based on the NRC (1998; 2012) nutrient requirements for broilers. ^2^ The premix provided the following per kg of diet: VA 12000 IU, VD_3_ 3200 IU, VE 34 IU, VK 1.0 mg, VB_1_ 2.4 mg, VB_2_ 5 mg, calcium pantothenate 15 mg, nicotinic acid 40 mg, pyridoxine 5 mg, biotin 0.25 mg, folic acid 1.4 mg, VB_12_ 0.02 mg, choline chloride 800 mg, manganese 70 mg, iodine 0.7 mg, iron 100 mg, cuprum 10 mg, zincum 80 mg, selenium 0.34 mg. ^3^ Calculated value (NRC 1998; 2012). ^4^ Analyzed content.

**Table 2 animals-13-03545-t002:** Effects of fermented feed on serum immune indexes of chickens at 22 d of age.

Items	Groups		*p*-Value
	CK	FJ	
IgA, g/L	1.91 ± 0.04 ^b^	2.69 ± 0.05 ^a^	<0.001
IgG, g/L	4.76 ± 0.10 ^b^	5.32 ± 0.10 ^a^	0.002
IgM, g/L	1.47 ± 0.07 ^b^	1.91 ± 0.08 ^a^	0.001
IL-1β, pg/mL	35.07 ± 0.49 ^a^	19.18 ± 0.43 ^b^	<0.001
IL-6, pg/mL	218 ± 6.21 ^a^	120 ± 4.34 ^b^	<0.001

Note: The different superscripts within a row indicate a significant difference (*p* < 0.05). IgA: Immunoglobin A; IgG: Immunoglobin G; IgM: Immunoglobin M; IL-1β: Interleukin-1β; IL-6: Interleukin-6.

**Table 3 animals-13-03545-t003:** Effects of fermented feed on digestive enzyme activities of chickens at 22 d of age.

Items	Groups		*p*-Value
	CK	FJ	
Duodenum			
Amylase, U/mg·protein	5.68 ± 1.2	4.77 ± 1.04	0.580
Trypsin, U/mg·protein	1536 ± 135.03 ^b^	2967 ± 345.6 ^a^	0.003
Lipase, U/g·protein	7.66 ± 1.38	8.55 ± 1.59	0.620
Jejunum			
Amylase, U/mg·protein	3.09 ± 0.88	4.39 ± 0.82	0.305
Trypsin, U/mg·protein	25977 ± 336.71	26507 ± 324.52	0.121
Lipase, U/g·protein	6.29 ± 1.25	25.82 ± 9.92	0.110
Ileum			
Amylase, U/mg·protein	2.63 ± 0.43	4.6 ± 1.21	0.190
Trypsin, U/mg·protein	25077 ± 175.96	25957 ± 347.06	0.238
Lipase, U/g·protein	21.85 ± 7.12	22.42 ± 7.69	0.568

Note: The different superscripts within a row indicate a significant difference (*p* < 0.05).

**Table 4 animals-13-03545-t004:** Effects of fermented feed on intestinal tissue morphology of chicks at 22 d of age.

Items	Groups		*p*-Value
	CK	FJ	
Duodenum			
Villus height, μm	954 ± 30.51	1026 ± 75.44	0.382
Crypt depth, μm	169 ± 15.58	141 ± 10.35	0.141
Villus height/Crypt depth, μm/μm	6.52 ± 0.75	8.07 ± 0.91	0.197
Jejunum			
Villus height, μm	794 ± 32.89	822 ± 43.64	0.620
Crypt depth, μm	121 ± 10.86	1162 ± 8.57	0.279
Villus height/Crypt depth, μm/μm	6.92 ± 0.7	7.08 ± 0.7	0.709
Ileum			
Villus height, μm	769 ± 18.57 ^a^	527 ± 31.56 ^b^	0.000
Crypt depth, μm	162 ± 13.64	132 ± 10.08	0.086
Villus height/Crypt depth, μm/μm	4.78 ± 0.40	4.23 ± 0.35	0.312

Note: The different superscripts within a row indicate a significant difference (*p* < 0.05).

**Table 5 animals-13-03545-t005:** Analysis of cecal microflora α diversity index of chickens at 22 d of age.

Items	Groups		*p*-Value
	CK	FJ	
Chao1 index	997 ± 32.71	10387 ± 39.96	0.437
Richness index	8397 ± 34.33	853 ± 29.86	0.767
Shannon index	5.94 ± 0.05 ^b^	6.32 ± 0.1 ^a^	0.040
Simpson index	0.08 ± 0.01	0.05 ± 0.01	0.098
ACE index	1002 ± 34.15	1045 ± 34.74	0.400
Evenness index	0.61 ± 0.004	0.64 ± 0.015	0.119

Note: The different superscripts within a row indicate a significant difference (*p* < 0.05). The Chao1 Index is an index that estimates the number of OTUs contained in a sample. The richness index is a measure of how many different species are present in a sample’s ecosystem. The Shannon index is used to estimate the microbial diversity in a sample, which takes into account the richness and evenness of the sample. The Simpson Index is a commonly used index to assess community diversity by calculating the probability that two randomly sampled sequences in a community belong to different species. The ACE index is an index that assesses the richness and evenness of species composition in a sample. The evenness index is an index that measures the degree of difference in the number of different species in a sample ecosystem.

## Data Availability

Data presented are original and not inappropriately selected, manipulated, enhanced, or fabricated.

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
