# Peer review of "The Effects of Fermented Feed on the Growth Performance, Antioxidant Activity, Immune Function, Intestinal Digestive Enzyme Activity, Morphology, and Microflora of Yellow-Feather Chickens"

_animals, 2023, doi:10.3390/ani13223545_

Round 1

Reviewer 1 Report

Comments and Suggestions for Authors

This study the author aimed to investigate the effects of fermented feed on the growth performance, antioxidant activity, immune function, intestinal digestive enzyme activity, morphology and microflora of yellow feather chickens, and offer a more theoretical basis for the exploitation and utilization of new fermented feed resources and the sustainable and healthy development of poultry farming. The author is correct about the development of this field, but does not have his own novel and unique academic views on the future development trend. There are several aspects of this article that need to be noted. I suggest that the author address the following issues.

1. The Sample summary section has been repeating the narration with no concrete meaning. It is recommended for the author to delete and rewrite it.

2. Abstract must contain most of the key information of the paper in brief form. It should conclude “research problem and objectives”,“methods”,“results or arguments”, and “conclusion”. It is recommended for the author to rewrite it.

3. Should there be a space between “P” and the following punctuation, as well as between punctuation and “value”? Please ensure consistency throughout the entire text.

4. Line 65-68,“Fermented feed has a promising future ……and replaceable to antibiotic. What evidence does the author provide to support this viewpoint?

5. Line 70-72,“In China, the rapid development of fermented products has made a significantly good effect in the practical production, though the start of fermented feed was late there (5-7)。” The author points out that the development of fermented feed in China started relatively late. However, the corresponding references provided are related studies from 2019 to 2022, but this is not the exact time when fermented feed started in China. Please provide a clear and explanatory explanation.

6. When making statements, the author should try to use concise short sentences. The conclusion section has only one sentence, making it difficult to understand.

7. There are too many tables in the article. It is recommended that the author create a graph or integrate tables of the same type.

8. The grammar of the entire article should be polished by someone whose native language is English

Comments on the Quality of English Language

Extensive editing of English language required

Author Response

Dear Reviewer 1#,

Thank you for comments concerning our manuscript entitled Effects of Fermented Feed on the Growth Performance, Antioxidant Activity, Immune Function, Intestinal Digestive Enzyme Activity, Morphology and Microflora of Yellow-Feather Chickens, ID: animals-2676545.

Comments: This study the author aimed to investigate the effects of fermented feed on the growth performance, antioxidant activity, immune function, intestinal digestive enzyme activity, morphology and microflora of yellow feather chickens, and offer a more theoretical basis for the exploitation and utilization of new fermented feed resources and the sustainable and healthy development of poultry farming. The author is correct about the development of this field, but does not have his own novel and unique academic views on the future development trend. There are several aspects of this article that need to be noted. I suggest that the author address the following issues.

Response: Thank you. Your comments are all valuable and very helpful for revising and improving our paper, as well as the important guiding significance to our researcher. We have studied comments carefully and have made a correction which we hope meet with approval. Revised portions are marked in Red in the paper.

Concern 1: The Sample summary section has been repeating the narration with no concrete meaning. It is recommended for the author to delete and rewrite it.

Response: Thank you. According to the requirements of Animals, we have readjusted the content of this section and modified the grammar.

Concern 2: Abstract must contain most of the key information of the paper in brief form. It should conclude “research problem and objectives”, “methods”, “results or arguments”, and “conclusion”. It is recommended for the author to rewrite it.

Response: Thank you. And we have revised it according to your request.

Concern 3: Should there be a space between “P” and the following punctuation, as well as between punctuation and “value”? Please ensure consistency throughout the entire text.

Response: Thank you. And we have revised it in the entire text.

Concern 4: Line 65-68,“Fermented feed has a promising future ……and replaceable to antibiotic.” What evidence does the author provide to support this viewpoint?

Response: Thank you. We added 3 references in this part.

  1. Missotten, J. A.; Michiels, J.; Dierick, N; Ovyn, A.; Akbarian, A.; De Smet, S. Effect of fermented moist feed on performance, gut bacteria and gut histo-morphology in broilers. Poult. Sci. 2013, 54, 627-34. doi: 10.1080/00071668.2013.811718
  2. Heres, L.; Engel, B.; van Knapen, F.; de Jong, M. C.; Wagenaar, J. A.; Urlings, H. A. Fermented liquid feed reduces susceptibility of broilers for Salmonella enteritidis. Sci. 2003, 82, 603-11. doi: 10.1093/ps/82.4.603
  3. Heres, L.; Engel, B.; Van Knapen, F.; Wagenaar, J. A.; Urlings, B. A. Effect of fermented feed on the susceptibility for Campylobacter jejuni colonisation in broiler chickens with and without concurrent inoculation of Salmonella enteritidis. Int J Food Microbiol. 2003, 87, 75-86. doi: 10.1016/s0168-1605(03)00055-2

Concern 5: Line 70-72,“In China, the rapid development of fermented products has made a significantly good effect in the practical production, though the start of fermented feed was late there (5-7)。” The author points out that the development of fermented feed in China started relatively late. However, the corresponding references provided are related studies from 2019 to 2022, but this is not the exact time when fermented feed started in China. Please provide a clear and explanatory explanation.

Response: Thank you. We have modified it in this part.

In recent years, the rapid development of fermented products has had a significantly good effect on animal husbandry production[10-12].

  1. Yan, J.; Zhou, B.; Xi, Y.; Huan, H.; Li, M.; Yu, J.; Zhu, H.; Dai, Z.; Ying, S.; Zhou, W.; et al. Fermented feed regulates growth performance and the cecal microbiota community in geese. Poult. Sci. 2019, 98, 4673-4684. doi: 10.3382/ps/pez169
  2. Wang, C.; Wei, S.; Xu, B.; Hao, L.; Su, W.; Jin, M.; Wang, Y. Bacillus subtilis and Enterococcus faecium co‐fermented feed regulates lactating sow's performance, immune status and gut microbiota. Microb. Biotechnol. 2021, 14, 614-627. doi: 10.1111/1751-7915.13672
  3. Zhu, Y.; Tao, Z.; Chen, X.; Xiao, J.; Zhang, Y.; Wang, Z. Effects of broussonetia papyrifera-fermented feed on production performance, egg quality, and caecal microbiota of laying hens during the late laying period. Ital. J. Anim. Sci. 2022, 21, 659-672. doi: 10.1080/1828051X.2022.2052368

Concern 6: When making statements, the author should try to use concise short sentences. The conclusion section has only one sentence, making it difficult to understand.

Response: Thank you. we have modified it.

In this study, the 2.00% fermented feed improved the growth performance, antioxi-dant activity, immune function, intestinal digestive enzyme activity, morphology, and microflora of yellow feather chickens.

Concern 7 There are too many tables in the article. It is recommended that the author create a graph or integrate tables of the same type.

Response: Thank you. We have change two tables to figures, and it has 5 tables and 6 figures in this manuscript.

Concern 8 The grammar of the entire article should be polished by someone whose native language is English.

Response: Thank you. We have modified the grammar of the entire article.

Special thanks to you for your good comments.

Hongzhi Wu,

[email protected]

Reviewer 2 Report

Comments and Suggestions for Authors

Dear authors, the manuscript ... accurately presents the utilization of a fermented product in chickens. The authors effectively outline the methodology and findings obtained. The discussion is appropriate.

I have only one observation. The statement "As the digestive system and immune system of chickens are not well developed" lacks sufficient support from the cited reference. Authors should include references that sufficiently substantiate this statement.

Author Response

Dear Reviewer 2#,

Thank you for comments concerning our manuscript entitled Effects of Fermented Feed on the Growth Performance, Antioxidant Activity, Immune Function, Intestinal Digestive Enzyme Activity, Morphology and Microflora of Yellow-Feather Chickens, ID: animals-2676545.

Comments: Dear authors, the manuscript ... accurately presents the utilization of a fermented product in chickens. The authors effectively outline the methodology and findings obtained. The discussion is appropriate.

Response: Thank you. Your comments are all valuable and very helpful for revising and improving our paper, as well as the important guiding significance to our researcher. We have studied comments carefully and have made a correction which we hope meet with approval.

Concern 1: I have only one observation. The statement "As the digestive system and immune system of chickens are not well developed" lacks sufficient support from the cited reference. Authors should include references that sufficiently substantiate this statement.

Response: Thank you. And we have added the corresponding references in this section..

Special thanks to you for your good comments.

Hongzhi Wu,

[email protected]

Reviewer 3 Report

Comments and Suggestions for Authors

Abstract: reduce to 200 words

In the text write the references in square brackets

Line 50: delete the highlighted part

Line 68: expand the concept by citing some work in which they were used

Line 106: specify sex and breed

Line 120: add a table in which analyze the composition of fermented feed

Line 138: specify catalogue number of all kits

Line 144: specify catalogue number of all kits

Line 146: reverse the order, first paraffin and then EE

Line 153: specify catalogue number of DNA Kit

Line 205: add a figure showing all EE of all tracts, highlighting how the measures were taken

Line 212: explain the significance of all indexes

all figures must be moved after they are first mentioned in the text

Reference: rewrite according journal’s instructions https://www.mdpi.com/journal/animals/instructions

Comments on the Quality of English Language

Minor editing of English language required

Author Response

Dear Reviewer 3#,

Thank you for comments concerning our manuscript entitled Effects of Fermented Feed on the Growth Performance, Antioxidant Activity, Immune Function, Intestinal Digestive Enzyme Activity, Morphology and Microflora of Yellow-Feather Chickens, ID: animals-2676545.

Concern 1: Abstract: reduce to 200 words

Response: Thank you. And we have revised it.

Concern 2: In the text write the references in square brackets

Response: Thank you. And we have modified them.

Concern 3: Line 50: delete the highlighted part

Response: Thank you. We have deleted it.

Concern 4: Line 68: expand the concept by citing some work in which they were used

Response: Thank you. And we have modified it and added 3 references.

Concern 5: Line 106: specify sex and breed

Response: Thank you. We have modified in the M&M parts.

Concern 6: Line 120: add a table in which analyze the composition of fermented feed

Response: Thank you. we have added it in table 1.

Concern 7: Line 138: specify catalogue number of all kits

Response: Thank you. We have modified it.

Concern 8: Line 144: specify catalogue number of all kits

Response: Thank you. We have modified it.

Concern 9: Line 146: reverse the order, first paraffin and then EE

Response: Thank you. We have modified it.

Concern 10: Line 153: specify catalogue number of DNA Kit

Response: Thank you. We have modified it.

Concern 11: Line 205: add a figure showing all EE of all tracts, highlighting how the measures were taken

Response: Thank you. We have modified it, the measure methods were shown in the M&M parts.

Concern 12: Line 212: explain the significance of all indexes

Response: Thank you. We have modified them.

Concern 13: all figures must be moved after they are first mentioned in the text

Response: Thank you. We have modified them.

Concern 14: Reference: rewrite according journal’s instructions

https://www.mdpi.com/journal/animals/instructions

Response: Thank you. We have modified all the references.

Special thanks to you for your good comments.

Hongzhi Wu,

[email protected]

Round 2

Reviewer 1 Report

Comments and Suggestions for Authors

Thank you very much!!